# 1,2-Propylene Glycol: A Biomarker of Exposure Specific to e-Cigarette Consumption

**Therese Burkhardt, Nikola Pluym, Gerhard Scherer and Max Scherer ***

ABF Analytisch-Biologisches Forschungslabor GmbH, Semmelweisstr. 5, 82152 Planegg, Germany;
therese.burkhardt@abf-lab.com (T.B.); nikola.pluym@abf-lab.com (N.P.); gerhard.scherer@abf-lab.com (G.S.)
* Correspondence: max.scherer@abf-lab.com

**Abstract:** Over the past decade, new emerging tobacco and nicotine-delivery products have changed the tobacco landscape. Especially, electronic cigarettes (ECs) have been suggested to be considered for tobacco harm reduction, reinforcing the need to identify novel biomarkers of exposure (BoE) specific to the EC use as this would complement exposure assessment and product compliance monitoring. Therefore, a sensitive LC-MS/MS method for the quantification of 1,2-propylene glycol (PG) and glycerol (G), the main e-liquid constituents, was established. PG and G were analyzed in plasma and urine samples from a clinical study comparing five nicotine product user groups, users of combustible cigarettes (CC), electronic cigarettes (EC), heated tobacco products (HTP), oral tobacco (OT), and oral/dermal nicotine delivery products (used for nicotine replacement therapy, NRT) with a control group of non-users (NU). Data demonstrate significantly elevated PG levels in urine and plasma in EC users compared to users of CC, HTP, NRT, OT as well as NU. In addition, PG in plasma and urine of vapers significantly correlated with nicotine (plasma) and total nicotine equivalents (urine), biomarkers reflecting product consumption, emphasizing the high specificity of PG as a BoE for EC consumption. We therefore suggest the use of PG as BoE in urine and/or plasma in order to monitor EC use compliance in exposure assessments.

**Keywords:** propylene glycol; electronic cigarette; biomarker of exposure; compliance marker



## 1. Introduction

Over decades, the measurement of biomarkers of exposure (BoE) has contributed important data to evaluate the health risk from cigarette smoking [1]. The two most common BoE that can be evaluated for all nicotine containing products are nicotine itself, as the most abundant alkaloid found in the tobacco leaf [2,3] as well as cotinine, its major metabolite possessing a longer half-life. Other relevant urinary biomarkers in tobacco smoke exposure assessment originate from tobacco-specific nitrosamines (TSNAs) (e.g., NNN, NNAL), polycyclic aromatic hydrocarbons (PAHs) (e.g., 1-hydroxypyrene, 3-hydroxybenzo[a]pyrene), aromatic amines (e.g., ortho-toluidine, 1-/2-naphthylamine, 3-/4-aminobiphenyl), and mercapturic acids of volatile organic compounds (VOCs) (e.g., 3-HPMA, CEMA) [2,4–9]. In the past decade, the tobacco landscape has changed, and a variety of new tobacco and nicotine-delivery products have been developed that pose a potentially reduced risk for the consumer as compared to smoking cigarettes. In 2015, Public Health England suggested to consider electronic cigarettes (ECs) for tobacco harm reduction, as complete switching could help reduce smoking related diseases [10]. They substantiated this claim in their most recent evidence update report in 2021 to which vaping of ECs is positively associated with successfully quitting smoking [11]. Still, a controversial debate about the benefits and risks of ECs continues to date [12]. Obviously, there is a need to identify BoE specific to EC consumption for a profound exposure and risk assessment [13–16]. However, to our knowledge, there is as yet no specific BoE to distinguish the use of ECs from the concomitant use of other tobacco/ nicotine products (dual or multiple product use).

E-liquids of ECs contain, in addition to nicotine, flavoring chemicals and carrier solvents, which are often referred to as humectants or stabilizing agents [17]. Mainly 1,2-propylene glycol (PG) and glycerol (G) are used as carrier solvents, constituting 80–95% of the e-liquid [17–19]. The vaporized PG and G generate an aerosol which is inhaled and thus absorbed by the user. Up to 45% of the unchanged PG and G are excreted in urine [20–22], thus becoming potential biomarker candidates for EC consumption. Schick et al. already discussed PG as appropriate BoE for EC consumption, but considered it as not suitable due to its widespread occurrence in daily use consumer products [5]. PG and G, as color- and odorless, water-soluble fluids, have beneficial properties as solvents, humectants, and antifreeze agents, making it attractive for a variety of applications across various industries such as food, pharmaceutical, cosmetic, medical, the manufacture of paints and coatings, and the production of plasticizers and polyester resins [17,23–26]. The widespread use of PG and G resulting in a general exposure of the population requires to verify, whether significant differences in the PG and G exposure are detectable between EC users (vapers) and users of other tobacco and nicotine-containing products as well as non-users [5]. The differentiation between user groups of different tobacco/nicotine products could provide a better understanding of the exposure pattern and the related health effects. In addition, BoE or BoE patterns specific for the use of an individual product, such as EC, would be useful to monitor product compliance (ideally the sole use of one product) rather than relying on self-reports, which is of particular importance for epidemiological studies [4].

For this purpose, a controlled clinical study was conducted comparing five tobacco/nicotine product user groups, namely, smokers of combustible cigarettes (CC), EC vapers, heated tobacco product (HTP) users, oral tobacco (OT) users, and users of oral/dermal nicotine delivery products (used for nicotine replacement therapy, NRT) as well as a control group of non-users (NU) [27]. Urine and plasma samples were analyzed for their PG and G content and statistical evaluation was performed across the different product user groups. Furthermore, PG and G levels were investigated for their association with the vaping intensity.

## 2. Materials and Methods

### 2.1. Study Design

A controlled, single-center, open label trial was conducted comparing five nicotine product user groups, namely exclusive users of CC, EC, HTP, OT, NRT with a control group of non-users (NU). Detailed information regarding the study design and the study population is described in Sibul et al. [27]. The study protocol has been approved by the ethics committee of the Medical Association Hamburg. Ten subjects per group were confined for 76 h (diet-control, exclusive use of one product), during which free, uncontrolled use of the products (own brand) was allowed. The amount of PG and G in the e-liquids consumed by the subjects ranged from of 50–55% for PG and 45–50% for G, respectively. This information is based on the manufacturer's specifications of the consumed liquids, or on the self-reported PG/G content of the e-liquid base if the liquids were self-mixed. Blood samples were collected at 7 a.m. and 5 p.m. on each day starting in the evening of day -1, when the subjects were admitted to the clinic. In total, 420 plasma samples were analyzed in this study. All urine voids were collected separately throughout the course of the clinical study. The total volume of each void was determined gravimetrically together with the time of void. Urine fractions were pooled to get 12 h urines (U0, U1/2, U3, U4/5, U6, U7/8; Figure 1). For PG/G analysis, six 12 h- urine pools of the 76-h stay of each subject were analyzed resulting in a total number of 360 urine samples.

B – blood draw (processed to receive plasma)
U – urine fraction

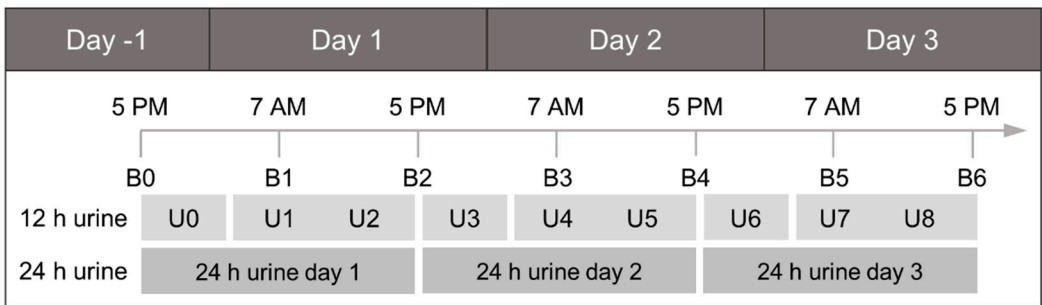

**Figure 1.** Time schedule for sample collection.

*2.2. Reagents and Chemicals*

Benzoyl chloride ($\geq$99%), glycine ($\geq$99%), and sodium hydroxide ($\geq$99%) were purchased from Merck KGaA (Darmstadt, Germany). n-Pentane ($\geq$99%) was supplied by VWR International GmbH (Darmstadt, Germany). Acetonitrile (min. 99.97%) was obtained from Th. Geyer GmbH & Co. KG (Renningen, Germany), bovine plasma from Biowest SAS (Nuaillé, France), and formic acid ($\geq$99%) from Biosolve (Dieuze, France).

Reference compounds 1,2-propylene glycol (99.9%) and glycerol (99.9%) were purchased from Sigma-Aldrich® a member of Merck KGaA (Darmstadt, Germany). Internal standards 1,2-propylene glycol-$d_6$ (99.6%) and glycerol-$d_5$ (98%) were obtained from CDN Isotopes Inc. (Quebec, QC, Canada).

*2.3. Analytical Method*

Urine and plasma samples were analyzed for their PG and G content according to Landmesser et al. [28]. An enzymatic hydrolysis experiment in urine using glucuronidase and sulfatase did not show an increase in PG and G concentration (data not shown), consequently this step was omitted for the sample preparation. In brief, 10 µL of an internal standard (IS) mixture containing 5 µg/mL 1,2-propylene glycol-$d_6$ and 5 µg/mL glycerol-$d_5$ were mixed with 25 µL of the urine or plasma sample. Derivatization was achieved by the addition of 500 µL of 4 M sodium hydroxide and 100 µL benzoyl chloride initiating the Schotten-Baumann reaction. n-Pentane (2 mL) was added and stirred for 15 min on a multi-tube vortex mixer. In order to quench the excess derivatization agent, 500 µL of a glycine solution in water (10% ($v/v$)) was added and subsequently mixed for another 15 min. Mixtures containing plasma were additionally precipitated at $<-70\ ^\circ$C for 15 min. After centrifugation of the sample (10 min, 1860 rcf), the supernatant was transferred into a new tube and evaporated to dryness using a vacuum concentrator. The sample was reconstituted in 100 µL of acetonitrile and analyzed by LC-MS/MS.

Analysis was performed by using an HTC PAL autosampler (CTC Analytics AG, Zwingen, Switzerland) with an Agilent 1100 HPLC system (Agilent Technologies, Inc., Waldbronn, Germany) hyphenated to an API 4000™ triple quadrupole mass spectrometer (MS/MS) (Sciex, Darmstadt, Germany). Analyst® Software (Version 1.5.3, Sciex, Framingham, United States) was used for data acquisition and quantification. A Kinetex® 5 µm EVO C18 (100 Å, 150 × 2.1 mm, Phenomenex Ltd., Aschaffenburg, Germany) equipped with a SecurityGuard™ ULTRA cartridge system for EVO-C18 (ID 2.1 mm Phenomenex Ltd., Aschaffenburg, Germany) as pre-column was used for chromatographic separation with an injection volume of 10 µL. The mobile phase consisted of 0.1% formic acid in water (A) and acetonitrile (B). Flow rate was set to 1.0 mL/min applying gradient elution as follows: initial conditions of 50% B were held for 0.6 min, increased to 60% B until 0.7 min, held at 60% B for 1.3 min, further increased to 80% B over the next 0.1 min, held for 1.9 min, increased to 95% B over 0.01 min, held for 1.99 min, decreased to 50% B within 0.01 min and held at 50% B for 1.99 min for re-equilibration, resulting in a total runtime of 8 min. Oven temperature was maintained at 40 °C. MS acquisition was carried out with an

electro spray ionization (ESI) ion source operated in positive ion mode. Source parameters were as follows: curtain gas: 30 psi, ion spray voltage: 5500 V, temperature: 300 °C, ion source gas 1: 60 psi, ion source gas 2: 50 psi, collision gas: 10 psi. The MS was operated in multiple reaction monitoring (MRM) mode with parameters specified in Table 1. Data were evaluated using Analyst® Software (Version 1.5.3, Sciex, Framingham, MA, USA) and Excel 2013 (Microsoft Cooperation, Redmond, WA, USA). Quadratic regression with 1/y weighting was applied.

**Table 1.** MS/MS parameter for 1,2-propylene glycol (PG) and glycerol (G).

| Analyte | Q1 m/z (Da) | Q3 m/z (Da) | | Declustering Potential (V) | Collision Energy (V) | Collision Cell Exit Potential (V) |
|---|---|---|---|---|---|---|
| 1,2-Propylene | 285 | 163 | Quantifier | 86 | 13 | 10 |
| glycol (PG) | 285 | 105 | Qualifier | 86 | 31 | 8 |
| Glycerol (G) | 405 | 283 | Quantifier | 91 | 13 | 16 |
| | 405 | 105 | Qualifier | 91 | 37 | 8 |
| 1,2-Propylene glycol-$d_6$ | 291 | 169 | IS for PG | 86 | 13 | 10 |
| Glycerol-$d_5$ | 410 | 288 | IS for G | 91 | 13 | 16 |

The determination of PG and G was performed in separate batches consisting of unknown samples, QC samples at low, medium, and high levels, calibrators, and blanks. Quantification of the study samples was performed by using water as surrogate matrix, as no analyte-free plasma or urine samples were available. Each calibration consisted of a blank, a zero, and eight non-zero concentration levels, including the LLOQ (lowest calibrator). Calibration ranged from 0.1 to 150 µg/mL. Deviation from the target values were evaluated for accepting calibrators and to verify the calibration range. The LLOQ was 0.1 µg/mL for PG and G in urine and plasma, respectively, and determined during method validation with a signal-to-noise ratio of at least 9 under consideration of the background levels, an accuracy of 80–120%, and a precision of ±20%. Human spot urine samples and bovine plasma spiked with the analytes to achieve three different concentration levels (low (L), medium (M), high (H)) reflecting the expected concentration range of the study samples were prepared as quality controls (QCs). For plasma QCs, 50 µg/mL PG and G (L) and 250 µg/mL PG and G (M, H) in water were added to bovine plasma so that the final QC concentrations were 0.4 µg/mL (L), 11.9 µg/mL (M), 115.5 µg/mL (H) for PG and 5.0 µg/mL (L) 15.4 µg/mL (M), 95.9 µg/mL (H) for G, respectively. For urinary QCs, 5 µg/mL G (L, PG present natively), 100 µg/mL (M) and 250 µg/mL (H) PG and G in water, respectively, were added to human urine so that the final QC concentrations were 0.7 µg/mL (L), 10.1 µg/mL (M), 123.7 µg/mL (H) for PG and 0.6 µg/mL (L), 9.8 µg/mL (M), 119.2 µg/mL (H) for G, respectively. More than 5% of unknown samples per analytical run (or at least 6 QC samples, two QC samples for each level) were randomly interspersed across the analytical runs as QC samples covering the expected range of analyte concentrations. This results in a total number of 27 QCs (9 per level) for PG and 24 QCs (8 per level) for G in urine and 27 QCs (9 per level) for PG and G in plasma, respectively.

In order to monitor the validity of the measurement, acceptance criteria as set forth in the FDA Guidance for Bioanalytical Method Validation [29] were used. Nicotine and 10 metabolites, namely cotinine, 3-OH-cotinine, nicotine glucuronide, cotinine glucuronide, 3-OH-cotinine glucuronide, 4-OH-4-(3-pyridyl)-butanoic acid, nornicotine, norcotinine, nicotine N-oxide, and cotinine N-oxide were determined by means of solid phase extraction and subsequent LC-MS/MS analysis in urine according to Piller et al. [30] in order to calculate the total nicotine equivalents (TNE).

### 2.4. Data Evaluation

PG and G values determined in urine or plasma with levels below LLOQ were reported as LLOQ/2. Urinary analyte concentrations (µg/mL) were multiplied with the respective 12 h urine volume to receive the amounts of PG and G excreted within 12 h. The appropriate 12 h urine pools were summed up to obtain the amount of the analytes

excreted in 24 h. Evening of the first day until evening of the next day were defined as a 24 h interval (U0 + U1/2, U3 + U4/5, U6 + U7/8). Urinary PG and G excretions were expressed in mg per 24 h (mg/24 h). Means, standard deviations (SD), and medians were calculated, where appropriate. Normal distribution was tested using Shapiro-Wilk and D'Agostino-K squared test. As PG and G concentrations were not normally distributed for the user groups investigated, the non-parametric Mann–Whitney U test (comparison of two groups) and Kruskal–Wallis-ANOVA (comparison of multiple groups) was used to investigate statistical significance between the different nicotine product user and non-user groups. Significance level was set to $\alpha = 0.01$. Correlations were evaluated with the non-parametric Spearman rank correlation analysis. Statistical analysis was conducted in OriginPro 2020b (Version 9.7.5.184, OriginLab Corporation, Northampton, MA, USA).

## 3. Results and Discussion

A robust method for the determination of PG and G in urine and plasma for human biomonitoring was established and fully validated according to FDA guidelines [29]. For PG, method accuracy rates were between 97.0–101.2% throughout the calibration range. Intra- and inter-day precisions were found to be <10% (CV) in urine (CV < 20% for levels < 3x LLOQ) and <8% (CV) in plasma (CV < 11% for levels < 3x LLOQ). In case of G, method accuracy rates were between 92.0–106.4% throughout the calibration range. Intra- and inter-day precisions were found to be <12% (CV) in urine and <15% (CV) in plasma. Carry-over was monitored by wash injections, whereby no contaminations above LLOQ were identified in this study. Quadratic calibration ranged from 0.1–150 µg/mL for PG and G in urine and plasma, respectively. For additional information with regard to method validation parameters see Supplementary Materials, Table S1.

PG and G were determined in 420 plasma and 360 urine samples of a clinical study. PG and G could be quantified above the LLOQ (0.1 µg/mL) in 360 (100%) and 342 (95%) of the urine and in 217 (52%) and 420 (100%) of the plasma samples, respectively. Representative chromatograms of PG from low and high concentrated urine and plasma samples are shown in Figure 2 (for G, see Figure S1 in Supplementary Materials).

Data within each group were comparable between different days, with exception for the PG level in the first plasma sample, when the subjects were admitted to the clinic (B0, day –1, 5 p.m.). Hence, data from day 3, i.e., the longest time period under confinement and thus under controlled conditions are most predictive of the product-use specific uptake, whereas results from day –1 are assumed to reflect the exposure under real-life conditions caused by the various sources of PG and G (for additional information with regard to day 1 and 2 see Supplementary Materials, Tables S2–S5). Consequently, slightly higher PG concentration in the B0 plasma sample in users of CC, HTP, NRT, OT, and NU compared with levels observed at the following sampling time points for these groups indicate that food and/or the use of daily care products might be a more important source for PG exposure than the use of CC, HTP, NRT and OT products (Figure 3).

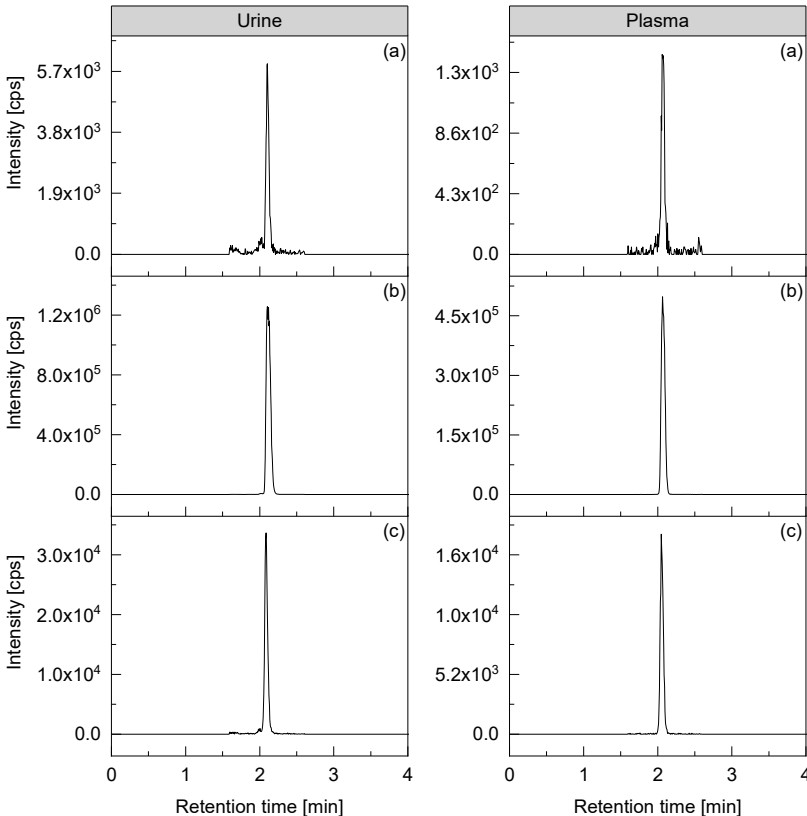

**Figure 2.** Representative chromatogram of 1,2-propylene glycol (PG) (MRM 285→163) of (**a**) low concentrated urine (0.23 μg/mL PG, user of oral tobacco) and plasma (0.13 μg/mL PG, user of nicotine replacement therapy) samples, (**b**) high concentrated urine (69.9 μg/mL PG, user of e-cigarettes) and plasma (31.9 μg/mL PG, user of e-cigarettes) samples, and (**c**) 1,2-propylene glycol-d$_6$ (MRM 291→169, 2 μg/mL).

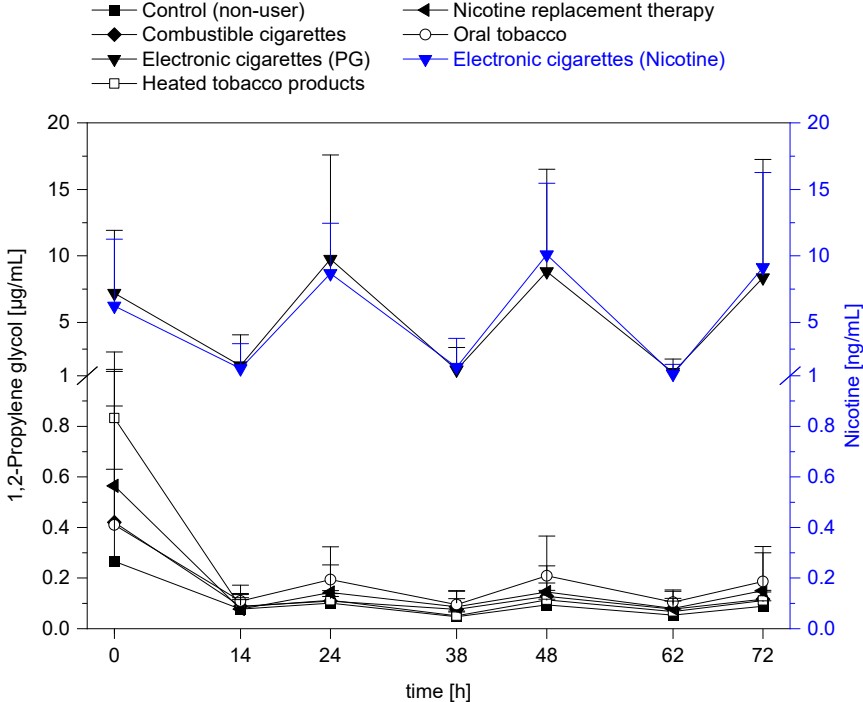

**Figure 3.** 1,2-Propylene glycol (PG) (μg/mL) as a function of time from day -1 to day 3 in plasma for the different nicotine product user groups and the control. Nicotine (ng/mL) in plasma is shown only for user of electronic cigarettes. Shown is the mean + standard deviation.

Data from analysis of G in plasma samples, with values ranging from $16.5 \pm 4.0$ µg/mL (mean $\pm$ SD) in OT to $21.5 \pm 4.6$ µg/mL in HTP, showed no significant differences ($p > 0.01$, Kruskal–Wallis-ANOVA) between the different groups investigated (Figure S2a, Table S2 in Supplementary Material) for both time points considered. In comparison, G level in urine were somewhat contradictory, as Kruskal–Wallis-ANOVA showed significant group separation ($p < 0.01$). However, closer examination revealed that the significant results (Mann–Whitney U test, $p < 0.01$) are due to approximately 1.5-fold lower G levels in EC ($2.8 \pm 0.6$ mg/24 h) compared to HTP ($4.1 \pm 1.1$ mg/24 h) users and users of NRT ($2.5 \pm 1.2$ mg/24 h) compared to HTP ($4.1 \pm 1.1$ mg/24 h), OT ($4.1 \pm 1.3$ mg/24 h), and NU ($3.8 \pm 1.2$ mg/24 h), respectively (Figure S2b, Table S2 in Supplementary Materials). In our opinion, the G levels are within the expected background exposure resulting from the daily uptake from food and consumer products [17,23–26] as well as G formed endogenously from proteins, pyruvate, glucose, triacylglycerols, and other glycerolipid metabolic pathways and excreted in urine [31–33]. The findings are in agreement with Landmesser et al. and Nelson et al., showing that oral intake of G below 0.05 g/kg body weight did not result in increased urinary glycerol excretion [28,34]. Data demonstrate that G is not elevated in users of EC, neither in plasma nor urine, and therefore cannot differentiate EC specific uptake of G from other nicotine products.

In contrast, PG levels in plasma showed significant differences ($p < 0.01$, Kruskal–Wallis-ANOVA) between the six groups investigated driven by the elevated levels in users of EC (Table 2, Figure 4a). PG levels in NU (control) ranged from $0.05 \pm 0.02$ µg/mL (7 a.m., day 3) to $0.09 \pm 0.02$ µg/mL (5 p.m., day 3) and were comparable to plasma PG levels observed in users of CC, HTP, NRT, and OT (Table 2). Results of the Kruskal–Wallis-ANOVA showed no significant differences between these groups ($p > 0.01$). These results indicate that there is only marginal exposure to PG when using CC, HTP, OT and NRT compared to the background exposure. In contrast, plasma PG levels in EC users were found to be 94-fold elevated in the evening (5 p.m.) and 24-fold elevated in the morning (7 a.m.) compared to NU and up to 71-fold at 5 p.m. and up to 18-fold at 7 a.m. compared to other nicotine product user groups (Table 2, Figure 4a). The observed increase of the PG levels in EC user was significant for both collection time points ($p < 0.01$, Mann–Whitney U test). No significant differences were found between the other groups investigated. Moreover, plasma PG increased significantly (Mann–Whitney U test, $p < 0.01$) throughout the day in EC users with average values of 1.25 µg/mL in the morning and 8.37 µg/mL in the evening (Figure 3). With the exception of the NU group, in which a significant increase of plasma PG from 0.05 µg/mL in the morning to 0.09 µg/mL in the evening was observed, none of the other groups showed an increase in PG levels over day.

**Table 2.** Descriptive statistics of 1,2-propylene glycol (PG) (µg/mL) in plasma on day 3 in the different nicotine user groups.

| User Group | Sampling | N Total | Mean | SD | Median | Min | Max |
|---|---|---|---|---|---|---|---|
| Control (non-user, NU) | 7 a.m. | 10 | 0.05 | 0.02 | 0.04 | 0.03 | 0.10 |
| | 5 p.m. | 10 | 0.09 *** | 0.02 | 0.09 | 0.05 | 0.12 |
| Combustible cigarettes (CC) | 7 a.m. | 10 | 0.08 | 0.04 | 0.06 | 0.04 | 0.17 |
| | 5 p.m. | 10 | 0.12 | 0.03 | 0.12 | 0.08 | 0.19 |
| Electronic Cigarettes (EC) | 7 a.m. | 10 | 1.25 | 1.01 | 0.78 | 0.25 | 2.95 |
| | 5 p.m. | 10 | 8.37 *** | 8.88 | 5.47 | 2.38 | 31.90 |
| Heated Tobacco Products (HTP) | 7 a.m. | 10 | 0.07 | 0.04 | 0.05 | 0.03 | 0.15 |
| | 5 p.m. | 10 | 0.11 | 0.04 | 0.11 | 0.05 | 0.17 |
| Nicotine Replacement Therapy (NRT) | 7 a.m. | 10 | 0.08 | 0.07 | 0.06 | 0.02 | 0.22 |
| | 5 p.m. | 10 | 0.15 | 0.17 | 0.08 | 0.04 | 0.59 |
| Oral Tobacco (OT) | 7 a.m. | 10 | 0.11 | 0.04 | 0.10 | 0.06 | 0.20 |
| | 5 p.m. | 10 | 0.19 | 0.12 | 0.19 | 0.07 | 0.43 |

SD: standard deviation, Min: minimum, Max: maximum. *** Statistically significant difference of PG between 7 a.m. and 5 p.m. (Mann–Whitney U test, $p < 0.01$).

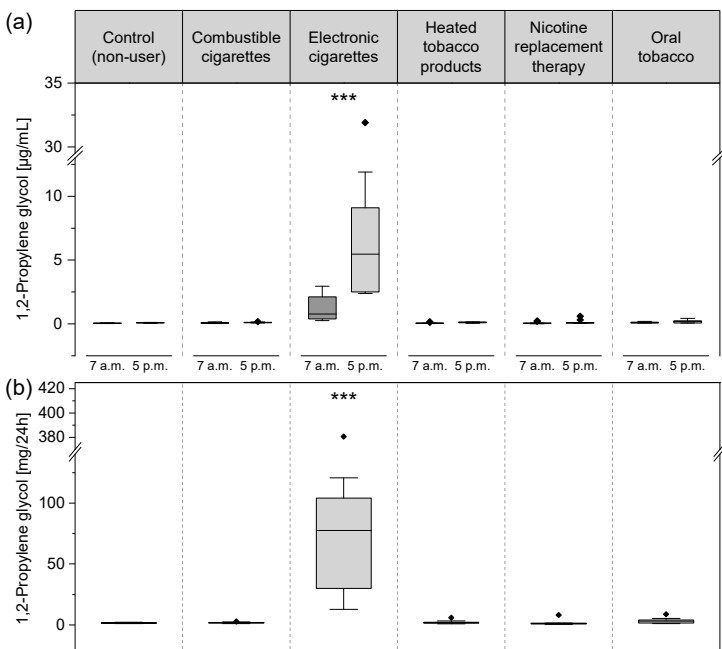

**Figure 4.** Box-and-whisker plots of 1,2-propylene glycol (PG) on day 3 in (**a**) plasma (µg/mL) at 7 a.m. and 5 p.m. and (**b**) urine (mg/24 h) of different nicotine product user groups and the control. Box-and-whisker plots represent medians (horizontal lines) with 25% and 75% percentiles (boxes), $1.5 \times$ IQR (whiskers), and outliers (diamonds). *** $p < 0.01$ (Mann–Whitney U test, comparison of EC with each other group).

Similar results were observed for PG in urine showing significantly elevated levels in EC users as compared to all other four nicotine user groups and the control group ($p < 0.01$, Kruskal–Wallis-ANOVA). PG in EC users showed levels of $95.4 \pm 107.1$ mg/24 h, 62 times higher than in NU (control) and between 29 to 54 times increased compared to the other nicotine user groups (Table 3, Figure 4b). Differences between EC and the other groups were statistically significant ($p = 1.1 \times 10^{-5}$, Mann–Whitney U test). The PG amount detected in the control group (NU) was in the range of $1.5 \pm 0.4$ mg/24 h (mean $\pm$ SD, Table 3, Figure 4b). No significant differences ($p < 0.01$, Mann–Whitney U test) were observed between the NU and users of CC ($1.8 \pm 0.5$ mg/24 h), HTP ($2.1 \pm 1.5$ mg/24 h), OT ($3.3 \pm 2.3$ mg/24 h), and NRT ($1.7 \pm 2.3$ mg/24 h). Kruskal–Wallis-ANOVA of these groups, excluding EC users, confirmed the results and showed no significant differences ($p > 0.01$).

**Table 3.** Descriptive statistics of 1,2-propylene glycol (PG) (mg/24 h) in urine on day 3 in the different nicotine user groups.

| User Group | N Total | Mean | SD | Median | Min | Max |
|---|---|---|---|---|---|---|
| Control (non-user, NU) | 10 | 1.5 | 0.4 | 1.5 | 1.0 | 2.1 |
| Combustible cigarettes (CC) | 10 | 1.8 | 0.5 | 1.7 | 1.0 | 2.7 |
| Electronic cigarettes (EC) | 10 | 95.4 *** | 107.1 | 77.6 | 12.7 | 380.7 |
| Heated tobacco products (HTP) | 10 | 2.1 | 1.5 | 1.6 | 0.9 | 5.9 |
| Nicotine replacement therapy (NRT) | 10 | 1.7 | 2.3 | 1.1 | 0.4 | 8.1 |
| Oral tobacco (OT) | 10 | 3.3 | 2.3 | 3.0 | 1.1 | 8.7 |

SD: standard deviation, Min: minimum, Max: maximum; ***: Statistically significant from all other groups ($p < 0.01$).

These results clearly demonstrate a separation between EC users and other nicotine product user groups in terms of PG levels in plasma and urine. Plasma PG concentrations and urinary PG excretion on day -1 (B0 and U0 samples) confirm these results, even though with higher variability, especially for users of HTP, NRT, OT (Figure S3 and Table S6 in Supplementary Materials). Also, under uncontrolled real-life conditions, PG levels in users of EC were significantly elevated compared with the other nicotine product user groups

and the control ($p < 0.01$, Mann–Whitney U test). PG levels in EC users were found to be increased 9- to 27-fold in plasma and 12- to 32-fold in urine samples, clearly demonstrating a group separation of EC users and other nicotine product user groups based on PG levels under uncontrolled conditions.

Although PG levels were only highly increased in urine and plasma samples of EC users, a general trend can also be observed for the other groups investigated: NU < CC, HTP, and NRT < OT << EC. The low PG levels detected in the control group (NU) in both urine and plasma samples, reflect the background concentrations, most probably from daily use of consumer products [17,23–26]. In relation to the excreted volume, PG levels in urine correspond to $672 \pm 209$ ng/mL, not significantly different (Mann–Whitney U test, $p > 0.01$) from Wurita et al. that observed PG levels of $5450 \pm 9290$ ng/mL (mean $\pm$ SD, range: 491–41600 ng/mL) in urine of 23 healthy subjects [35]. In addition, if PG was normalized to creatinine (determined as part of the project), the PG level in the control group (NU) was $1.6 \pm 0.6$ mmol/mol creatinine, which is comparable with the value of $2.3 \pm 1.4$ mmol/mol creatinine observed by Laitinen et al. [36]. A statement regarding plasma levels is not possible, as data on the general background exposure to PG in plasma are lacking in the literature. Slightly higher PG concentrations detected in users of CC, HTP, and NRT could be attributed to the addition of PG as humectant to conventional cigarettes [37–40], HTP [41,42], and nicotine sprays (e.g., Nicorette® Mint Spray) used for NRT. In comparison, even higher PG levels were observed in users of OT such as snus products, to which PG is also added as a humectant [43]. As mentioned before, the EC users show by far the highest PG values, which can be attributed to PG intake from EC use [44,45]. High variations in the group of EC users in urine and plasma can be ascribed to various factors affecting the PG uptake, such as varying PG contents in the used e-liquids, e-cigarette device characteristics such as model, wattage, and temperature, as well as the use behavior of the individual subject including puff number, puff volume, puff duration and depth of inhalation [46–55]. In fact, measured PG level highly depend on the PG content of the liquid consumed. However, as the plasma and urine PG levels were >9-fold elevated in EC users, we assume that the critical level to differentiate between EC user and other consumer groups would be a PG content <10%. Theoretical mathematical assessment with 5 times lower PG level in EC users still holds statistical significance substantiating the limit 10% PG content. We were able to demonstrate that the EC specific PG uptake is better reflected by its plasma concentrations measured at different time points, which showed a 6.7-fold increase throughout the day. Lower plasma PG levels in the morning can be attributed to the prohibition of EC use from midnight to 9 a.m. applied in the clinical study, leading to a partial washout within this time frame (Figure 3). The increase in plasma PG concentrations during the day was already observed by Landmesser et al. and can be unequivocally attributed to EC use [28]. The correlation of excreted PG with the actual amount of PG inhaled by the individual user would be of utmost interest, as it would contribute to the comprehensive assessment of PG intake after EC use [56,57]. Therefore, the PG intake was calculated as an estimate according to Equation (1), where $I_{PG}$ is the intake of PG (g/d) from EC use, LPD is the amount of e-liquid (g) consumed per day, MSp is the mouth spill, and R is the respiratory retention. As a first approximation, a PG content of 50%, a mouth spill of 30% [58], and a respiratory retention of 92% [59] was assumed.

$$I_{PG} = LPD \times 0.5 \times (1 - MSP) \times R \qquad (1)$$

The urinary PG excretion (mg/24 h) showed a significant ($p < 0.01$) correlation with the calculated PG intake with a Spearman's rank correlation coefficient of r = 0.976 (Figure 5a). Additionally, the association of vaping-related PG uptake was indirectly addressed by correlating the PG biomarker levels in EC users with specific biomarkers that provide a measure of product consumption. Urinary PG measured on day 3 was compared with TNE, the molar sum of nicotine and 10 metabolites [60]. As expected, the urinary PG concentrations showed a correlation with the excreted amount of TNE with a Spearman's rank correlation coefficient of r = 0.66 ($p = 0.038$, Figure 5b) indicating that more PG is

absorbed and excreted upon higher uptake of EC aerosol. Plasma PG levels correlate only moderately with nicotine plasma levels on day 3 for both collection time points (7 a.m. and 5 p.m.), with a Spearman's rank correlation coefficient of r = 0.37 for each (Figure 5c). Although the correlation of plasma PG with nicotine is not significant, a clear relationship can be observed when considering both collection time points each (Figure 5c) and the plasma time course (Figure 3). This correlation could be explained by comparable plasma half-lives of nicotine (approx. 2 h after inhalation [61–63]) and PG (2.3 ± 0.7 h after intravenous infusion [64] and 3.8 ± 0.8 h/4.1 ± 0.7 h after oral administration [65], and approx. 2 h after vaping [28]), confirming previous findings [28].

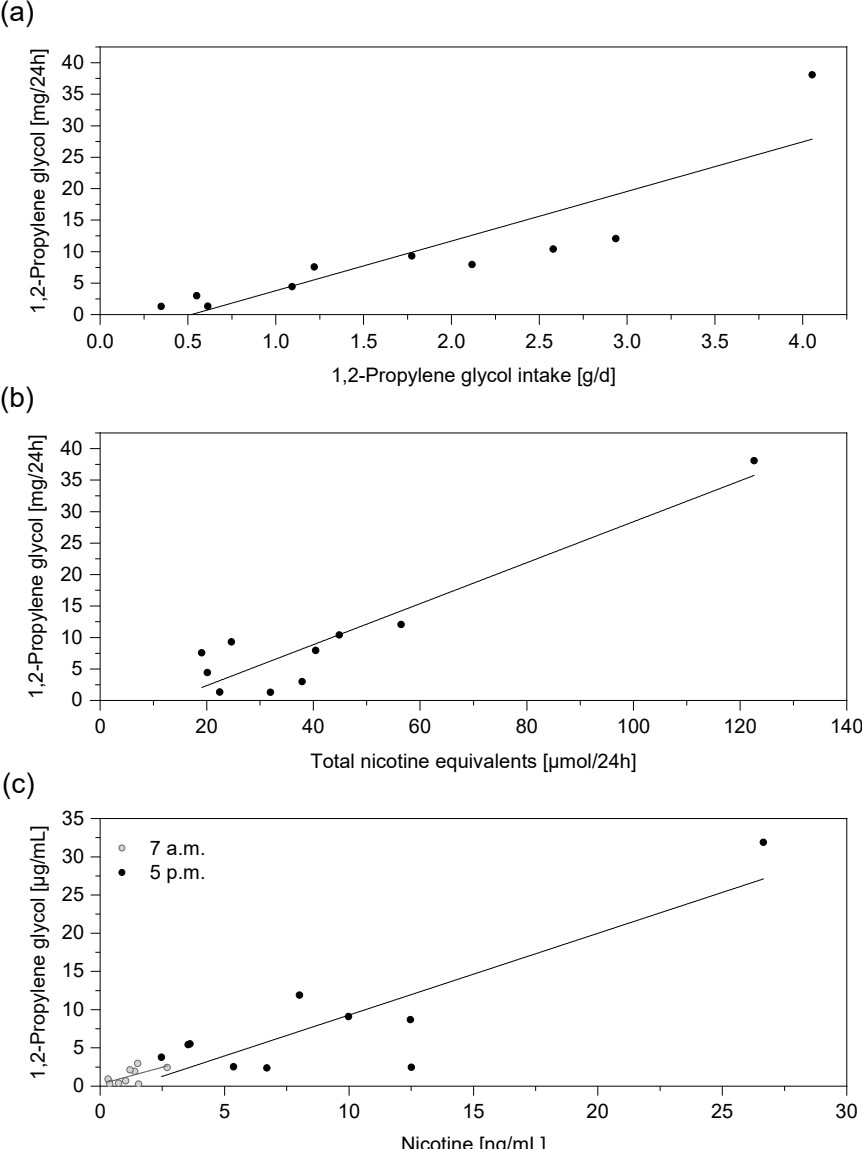

**Figure 5.** (**a**) Correlation between 1,2-propylene glycol (PG) in urine (mg/24 h) and the estimated PG intake (g/d) of EC users on day 3 (Spearman's correlation: r = 0.976 ($p = 1.4 \times 10^{-6}$), (**b**) correlation between PG in urine (mg/24 h) and urinary nicotine equivalents (μmol/24 h) of EC users (Spearman's correlation r = 0.661, $p = 0.0376$), (**c**) correlation of PG (μg/mL) and nicotine (ng/mL) in plasma of EC users on day 3 at 7 a.m. with a Spearman's correlation r = 0.370 ($p = 0.293$), and at 5 p.m. with a Spearman's correlation r = 0.370 ($p = 0.293$).

The daily PG intake of the EC users was in average 1.7 g PG/d (range: 0.3–4.1 g/d, estimated with Equation (1)). Mean plasma PG concentrations on the 3 study days at 5 p.m. were in the range of 8.4–9.8 μg/mL. These data are roughly comparable with a

study from Speth et al., in which a PG dose of 5.1 g was administered intravenously (IV) to 3 subjects over a time period of 4 h [64]. PG plasma concentrations were reported to be in the range of 48–60 µg/mL. While the dose in the IV study was on average three times higher compared to our study, the plasma levels were about 6-fold increased. This apparent discrepancy is best explained by the time period of dosing: 4 h in the IV study versus ~8 h in the present study. Interestingly, Speth et al. [64] reported a saturable PG clearance over the applied dose range of 5–21 g/d. Whether this phenomenon is also relevant of the vaping-related doses remains to be investigated. Finally, it is noteworthy that these authors found no evidence for lactate acidosis, hemolysis or increase in osmolality in the studied dose range [64].

An obstacle for using PG (in plasma or urine) as a biomarker of exposure might be the inter-individual variability in pharmacokinetic parameters such as half-life, volume of distribution and achievable plasma levels as reported for humans in the literature [64,65]. However, the strong dose-response relationships observed for urinary excretion of PG in vapers (Figure 5a,b) indicate that this is unlikely to be a major issue.

Although the emergence of EC in the last decade led to the need to identify a biomarker specific for EC use [5], most studies have focused on biomarkers of tobacco-smoke exposure, which are not specific to EC use. Moreover, studies often only comprise the comparison of EC to NU and/or CC alone [5,66–71]. Only a few studies include other nicotine product user groups [72,73]. To the best of our knowledge, this is the first study systematically assessing the PG and G concentration in plasma and urine samples in users of EC and four additional nicotine product user groups, namely users of CC, HTP, NRT, and OT. G levels did not differ between the different groups investigated, neither in urine nor in plasma. The reason for this is the high and variable background level of G in plasma and urine caused by a common exposure to G (food, consumer products) and the endogenous formation of G in the lipid metabolism. Therefore, G is not suitable as a specific biomarker to identify EC use. In contrast, levels of PG were significantly elevated in users of EC compared to the control group and all nicotine user groups investigated, despite the small sample size (10 subject per group), which represents a limitation of the current study. Larger studies under field conditions are required to support the suitability of PG in plasma or urine as specific biomarker for the use of ECs. Another limitation is a minimum amount of presumable $\geq 10\%$ PG in the e-liquid, to detect a difference between users of EC and other nicotine user groups. However, further investigations are needed in order to verify this cut-off.

## 4. Conclusions

In conclusion, the current study clearly demonstrates a significant distinction between users of EC and users of CC, HTP, NRT, OT as well as NU based on the PG level in urine and plasma. The observed dose-response relationship between urinary and plasma PG and intensity of vaping (daily consumption and nicotine uptake) emphasizes the suitability of PG as a potential biomarker of EC use. Due to the restricted sample size of the current study, we recommend verifying these results under field conditions. Consequently, we propose the use of PG in urine and/or plasma in order to monitor EC use compliance in exposure assessments under real-life conditions (field and epidemiological studies). Moreover, a combination of several biomarkers may lead to a more comprehensive differentiation among several user groups which would provide a better understanding of exposure and related health effects.

**Supplementary Materials:** The following are available online at https://www.mdpi.com/article/10.3390/separations8100180/s1, Table S1. Method validation parameters for the determination of propylene glycol and glycerol in urine and plasma, Figure S1. Representative chromatogram of glycerol (G) (MRM 405→283) of (a) low concentrated urine (0.21 µg/mL G, user of oral tobacco) and plasma (9.9 µg/mL G, non-user) samples, (b) high concentrated urine (9.7 µg/mL G, user of e-cigarettes) and plasma (24.4 µg/mL PG, user of e-cigarettes) samples, and (c) glycerol-d5 (MRM 410→288, 2 µg/mL), Figure S2. Box-and-whisker plots of glycerol (G) on day 3 in (a) plasma (µg/mL)

at 7 a.m. and 5 p.m. and (b) urine (mg/24 h) between different nicotine product user groups and the control. Box-and-whisker plots represent medians (horizontal lines) with 25% and 75% percentiles (boxes), 1.5xIQR (whiskers), and outliers (diamonds), Table S2. Descriptive statistics of glycerol (µg/mL) in plasma of study samples for different nicotine user groups, Table S3. Descriptive statistics of glycerol (mg/24 h) in urine of study samples for different nicotine user groups, Table S4. Descriptive statistics of 1,2-propylene glycol (µg/mL) in plasma of study samples for different nicotine user groups, Table S5. Descriptive statistics of PG (mg/24 h) in urine of study samples for different nicotine user groups, Figure S3. Box-and-whisker plot of 1,2-propylene glycol (PG) on day –1 in (a) plasma (µg/mL) at 5 p.m. (B0) and (b) urine (mg/12 h) (U0) between different nicotine product user groups and the control. Box-and-whisker plots represent medians (horizontal lines) with 25% and 75% percentiles (boxes), 1.5xIQR (whiskers), and outliers (diamonds). *** $p < 0.01$ (comparison of EC vs all other groups), Table S6. Descriptive statistics of PG in urine (mg/12 h) and plasma (µg/mL) of study samples for different nicotine user groups at day -1.

**Author Contributions:** Conceptualization, M.S., G.S. and N.P.; formal analysis, T.B.; writing— original draft preparation, T.B.; writing—review and editing, N.P., M.S. and G.S.; supervision, M.S.; project administration, M.S.; funding acquisition, M.S. and N.P. All authors have read and agreed to the published version of the manuscript.

**Funding:** This study was funded with a grant from the Foundation for a Smoke-Free World, a US nonprofit 501(c)(3) private foundation with a mission to end smoking in this generation. The Foundation accepts charitable gifts from PMI Global Services Inc. (PMI); under the Foundation's Bylaws and Pledge Agreement with PMI, the Foundation is independent from PMI and the tobacco industry. The contents, selection, and presentation of facts, as well as any opinions expressed herein are the sole responsibility of the authors and under no circumstances shall be regarded as reflecting the positions of the Foundation for a Smoke-Free World, Inc.

**Institutional Review Board Statement:** The study was conducted according to the guidelines of the Declaration of Helsinki, and was approved by the Ethics Committee of the Medical Association Hamburg (reference number: PV7084, date of approval: 10 September 2019).

**Informed Consent Statement:** Informed consent was obtained from all subjects involved in the study.

**Data Availability Statement:** The data presented here are available upon request from the corresponding author.

**Acknowledgments:** The authors thank CTC North GmbH (Hamburg, Germany) for conducting the clinical study, and our colleagues from ABF for their assistance in organizing and managing the clinical study, and performing the sample analysis.

**Conflicts of Interest:** The authors declare no conflict of interest. The funders had no role in the design of the study; in the collection, analyses, or interpretation of data; in the writing of the manuscript, or in the decision to publish the results.

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
