# Peer review of "1,2-Propylene Glycol: A Biomarker of Exposure Specific to e-Cigarette Consumption"

_separations, doi:10.3390/separations8100180_

Round 1

Reviewer 1 Report

The manuscript titled “1,2-Propylene glycol: A biomarker of exposure specific to e-cigarette consumption” is a tobacco industry-funded study that presents data using the e-cigarette solvents propylene glycol (PG) and glycerol (G) as biomarkers for e-cigarette (EC) use. The sample group is small (n=10), important information such as the used e-cigarettes and the PG/G ratio contained therein and use frequency is not reported, and the followed subjects were “confined” (not further elucidated), suggesting that the study conditions did not correspond to a real-life situation. Taken together, the results suggest that it is unlikely that looking at PG in urine or blood may be a sufficient biomarker to determine, for example in a clinical setting, if a person is a current EC user. Given that the trend is clearly towards more glycerol-heavy e-cigarettes, for example, if a user used a glycerol-heavy e-cigarette such as Juul (30/70 PG/G) or a >80% G e-liquid as is common in 3rd generation devices, these users would likely be classified as non-EC-users by the investigated method. Especially the data shown for Day0 fortifies this as it is clear that users that were not “confined” did show large PG backgrounds (data only shown for urine). In addition, it is unlikely that the sample group 10 was sufficient to draw such large conclusions. It is thus recommended to reject the manuscript in its current format, as the conclusions are not supported by the results.

Author Response

Response to reviewers

Ref: separations-1360630

Title: 1,2-Propylene glycol: A biomarker of exposure specific to e-cigarette consumption

Journal: Separations

First of all, we would like to thank the reviewers for their helpful comments, which substantially improved the manuscript. We hope that we could address all raised issues. For better traceability we highlighted the added/revised text in red.

Reviewer 1:

The manuscript titled “1,2-Propylene glycol: A biomarker of exposure specific to e-cigarette consumption” is a tobacco industry-funded study that presents data using the e-cigarette solvents propylene glycol (PG) and glycerol (G) as biomarkers for e-cigarette (EC) use. The sample group is small (n=10), important information such as the used e-cigarettes and the PG/G ratio contained therein and use frequency is not reported, and the followed subjects were “confined” (not further elucidated), suggesting that the study conditions did not correspond to a real-life situation. Taken together, the results suggest that it is unlikely that looking at PG in urine or blood may be a sufficient biomarker to determine, for example in a clinical setting, if a person is a current EC user. Given that the trend is clearly towards more glycerol-heavy e-cigarettes, for example, if a user used a glycerol-heavy e-cigarette such as Juul (30/70 PG/G) or a >80% G e-liquid as is common in 3rd generation devices, these users would likely be classified as non-EC-users by the investigated method. Especially the data shown for Day0 fortifies this as it is clear that users that were not “confined” did show large PG backgrounds (data only shown for urine). In addition, it is unlikely that the sample group 10 was sufficient to draw such large conclusions. It is thus recommended to reject the manuscript in its current format, as the conclusions are not supported by the results.

Response:

First of all, we would like to thank the reviewer for the comprehensive review of the manuscript.

We agree with the reviewer that the sample size of the study (N = 10) is quite small and kindly refer to section 3 (Results and Discussion) where this is listed as limitation of the current study. For this reason, we have decided to use a fairly hard cut-off for significance, α = 0.01, to address this limitation. We further suggest to verify these results in a larger study under field conditions and have added a sentence in section 4 (Conclusion), in addition to section 3 (Results and Discussion), to indicate this.

Section 3. Results and Discussion. “In contrast, levels of PG were significantly elevated in users of EC compared to the control group and all nicotine user groups investigated, despite the small sample size (10 subject per group), which represents a limitation of the current study. Larger studies under field conditions are required to support the suitability of PG in plasma or urine as specific biomarker for the use of ECs.”

Section 4. Conclusion.

“Due to the restricted sample size of the current study, we recommend verifying these re-sults under field conditions.”

To address the PG/G content, additional information on the PG/G content of the e-liquids consumed by the subjects has been added in section 2.1 (Study design).

“The amount of PG and G in the e-liquids consumed by the subjects ranged from of 50-55 % for PG and 45-50 % for G, respectively.”

We have tried to keep the section on the study design as brief as possible, since detailed information is given in Sibul et al. (2021). Nevertheless, we have added some information on the confinement.

“Ten subjects per group were confined for 76 h (diet-control, exclusive use of one product), during which free, uncontrolled use of the products (own brand) was allowed.”

We agree with the reviewer that evaluation of day three PG data alone does not provide information on real-life conditions. For this reason, data from day -1 were assessed. Immediately after admission to the clinical site (day ‑1, 5 pm,) plasma and urine (12 h urine) samples were collected. These samples can be defined as real-life samples as the subjects were not subject to any restrictions prior to admission to the clinic. We’ve added Table S6 (Supplementary Material) showing the descriptive statistics of PG in plasma and urine on day -1 (U0, B0). In addition, Figure S3 in Supplementary Material shows urinary and plasma PG data on day -1.

Even in these uncontrolled real-life study samples, levels of PG were significantly elevated in users of EC compared to the control group and all nicotine user groups investigated. To substantiate this, we’ve provided you with a table showing the p-values of the significance test (Mann-Whitney U test) performed at day -1 for U0 (Table 1) and B0 (Table 2).

For day -1 we observed 9-27 times higher PG levels in plasma (B0) and 12-32 higher PG levels in urine (U0) in EC user compared to the other groups. Study subjects used e-liquids with an average PG content of 50 %. Even if the e-liquid PG content would decrease to 10 %, we would most likely see a difference between user of EC and other nicotine user groups.

To highlight the PG content of the e-liquid as a prerequisite for the differentiation, we have added this limitation to the discussion.   

“In fact, the measured PG levels highly depend on the PG content of the liquid consumed. However, since the plasma and urine PG levels were > 9-fold elevated in EC users, we assume that the critical level to differentiate between EC user and other consumer groups would be a PG content < 10 %.”

“Another limitation is a minimum amount of presumable ≥ 10 % PG in the e-liquid, to de-tect a difference between users of EC and other nicotine user groups. However, further investigations are needed in order to verify this cut-off.”

Table 1: p-values (Mann-Whitney U test) for the comparison of the urine fraction on day -1 (U0) of the different groups investigated in the clinical study. p-values < 0.01 are marked in red.

NU

CC

EC

HTP

NRT

OT

NU

--

1.0E+00

1.1E-05

1.1E-01

9.1E-01

3.9E-01

CC

1.0E+00

--

4.3E-05

7.5E-02

9.7E-01

3.2E-01

EC

1.1E-05

4.3E-05

--

1.3E-04

1.1E-05

2.2E-05

HTP

1.1E-01

7.5E-02

1.3E-04

--

3.9E-01

6.8E-01

NRT

9.1E-01

9.7E-01

1.1E-05

3.9E-01

--

3.2E-01

OT

3.9E-01

3.2E-01

2.2E-05

6.8E-01

3.2E-01

--

Table 2: p-values (Mann-Whitney U test) for the comparison of the plasma fraction on day -1 (B0) of the different groups investigated in the clinical study. p-values < 0.01 are marked in red.

NU

CC

EC

HTP

NRT

OT

NU

--

3.5E-02

1.1E-05

1.1E-01

5.8E-01

5.2E-02

CC

3.5E-02

--

1.1E-05

7.5E-01

1.1E-01

3.4E-01

EC

1.1E-05

1.1E-05

--

1.1E-05

1.1E-05

1.1E-05

HTP

1.1E-01

7.5E-01

1.1E-05

--

1.7E-01

1.7E-01

NRT

5.8E-01

1.1E-01

1.1E-05

1.7E-01

--

2.8E-01

OT

5.2E-02

3.4E-01

1.1E-05

1.7E-01

2.8E-01

--

Reviewer 2 Report

General comment:

The present work (separations-1360630) aims to evaluate the use of PG and G as biomarkers of exposure to e-cigarette consumption. Overall, I find the paper interesting and original, although there are several works dealing with this particular subject. The information described can be very interesting to tobacco researchers, yet Separations journal is more devoted to an analytical chemistry audience. Regarding the manuscript itself, it’s clear that the authors conducted their research in an appropriate manner and provided sufficient data that support their conclusions. The quality of written English is good. Specific comments are given in the pdf file attached.

Author Response

Response to reviewers

Ref: separations-1360630

Title: 1,2-Propylene glycol: A biomarker of exposure specific to e-cigarette consumption

Journal: Separations

First of all, we would like to thank the reviewers for their helpful comments, which substantially improved the manuscript. We hope that we could address all raised issues. For better traceability we highlighted the added/revised text in red.

Reviewer 2:

The present work (separations-1360630) aims to evaluate the use of PG and G as biomarkers of exposure to e-cigarette consumption. Overall, I find the paper interesting and original, although there are several works dealing with this particular subject. The information described can be very interesting to tobacco researchers, yet Separations journal is more devoted to an analytical chemistry audience. Regarding the manuscript itself, it’s clear that the authors conducted their research in an appropriate manner and provided sufficient data that support their conclusions. The quality of written English is good. Specific comments are given in the pdf file attached.

Response:

We agree with the reviewer that the information described can be very interesting to tobacco researchers and that the Journal Separations is more devoted to an analytical chemistry audience. Since our manuscripts combines both aspects we think that that it would be a valuable contribution to the Special Issue "Method Development and Applications for Reduced-Risk Products”.

Comment:

Page 1, line 30. I suggest rewriting this sentence. Remove "among others" and say "The two most common BoE..." The fact is that nicotine and cotinine are the two most common BoE and that's the most important info to retain. The next sentence already refers to other BoE that can be analyzed.

Response:

Thank you for your comment. We rephrased the sentence as suggested.

“The two most common BoE that can be evaluated for all nicotine containing products are nicotine itself, as the most abundant alkaloid found in the tobacco leaf [2,7] as well as cotinine, its major metabolite possessing a longer half-life.”

Comment:

Page 4, lines 143-146. Write down exactly how the spiking procedure was performed and clearly specify the PG and G concentrations at the low, mid and high levels. The FDA guideline specifies that besides a low, medium and high level, a level correspondent to the LLOQ should be used. Where is this info? Do the authors use this LLOQ level?

Response:

We have added the spiking procedure as well as spiking level for PG and G in urine and plasma (see section 2.3). We agree that the FDA guideline (2018) specifies the use of the LLOQ as additional QC level for validation. However, for analytical runs three QC levels (low, medium and high) are required that cover the expected study sample concentration range. For this reason, we did not include an additional QC level. Nevertheless, we tried to prepare QCL to be in the range of LLOQ. However, due to the background PG in the urine (coming from various sources) used for the preparation of the QCs, the PG level in QCL was slightly higher.

“For plasma QCs, 50 µg/mL PG and G (L) and 250 µg/mL PG and G (M, H) in water were added to bovine plasma so that the final QC concentrations were 0.4 µg/mL (L), 11.9 µg/mL (M), 115.5 µg/mL (H) for PG and 5.0 µg/mL (L) 15.4 µg/mL (M), 95.9 µg/mL (H) for G, respectively. For urinary QCs, 5 µg/mL G (L, PG present natively) , 100 µg/mL (M) and 250 µg/mL (H) PG and G in water, respectively, were added to human urine so that the final QC concentrations were 0.7 µg/mL (L), 10.07 µg/mL (M), 123.67 µg/mL (H) for PG and 0.6 µg/mL (L), 9.8 µg/mL (M), 119.17 µg/mL (H) for G, respectively.”

Comment:

Page 4, line 149. Describe in detail what parameters were under evaluation under the FDA guidance document and how they were evaluated. For example, how many calibrators were used? What was the tested range? How was the LLOQ determined? What are the LLOQ for PG and G? How many QCs replicates were used to evaluate the method's accuracy and precision?

Response:  

We’ve added information on the suggested parameters accordingly.

“The determination of PG and G was performed in separate batches consisting of unknown samples, QC samples at low, medium, and high levels, calibrators, and blanks. Quantification of the study samples was performed by using water as surrogate matrix, since no analyte-free plasma or urine samples were available. Each calibration consisted of a blank, a zero, and eight non-zero concentration levels, including the LLOQ (lowest calibrator). Calibration ranged from 0.1 to 150 µg/mL. Deviation from the target values were evaluated for accepting calibrators and to verify the calibration range. The LLOQ was 0.1 µg/mL for PG and G in urine and plasma, respectively, and determined during method validation with a signal-to-noise ratio of at least 9 under consideration of the background levels, an accuracy of 80-120 %, and a precision of ± 20 %.

Human spot urine samples and bovine plasma spiked with the analytes to achieve three different concentration levels (low (L), medium (M), high (H)) reflecting the expected concentration range of the study samples were prepared as quality controls (QCs). For plasma QCs, 50 µg/mL PG and G (L) and 250 µg/mL PG and G (M, H) in water, respectively, were added to bovine plasma so that the final QC concentrations were 0.4 µg/mL (L), 11.9 µg/mL (M), 115.5 µg/mL (H) for PG and 5.0 µg/mL (L) 15.4 µg/mL (M), 95.9 µg/mL (H) for G, respectively. For urinary QCs, 5 µg/mL G (L, PG present natively), 100 µg/mL (M) and 250 µg/mL (H) PG and G in water, respectively, were added to human urine so that the final QC concentrations were 0.7 µg/mL (L), 10.07 µg/mL (M), 123.67 µg/mL (H) for PG and 0.6 µg/mL (L), 9.8 µg/mL (M), 119.17 µg/mL (H) for G, respectively. More than 5 % of unknown samples per analytical run (or at least 6 QC samples, two QC samples for each level) were randomly interspersed across the analytical runs as QC samples covering the expected range of analyte concentrations. This results in a total number of 27 QCs (9 per level) for PG and 24 QCs (8 per level) for G in urine and 27 QCs (9 per level) for PG and G in plasma, respectively.”

Comment:

Page 4, lines 150-151. This is already a result of this study. Move this information to the results section. In addition, what is the LLOQ for PG and G? I can't find this info.

Response:

This information was moved to section 3 (Results and Discussion). Information regarding the LLOQ has been added to the section 2.3 (Analytical Method).

“The LLOQ was 0.1 µg/mL for PG and G in urine and plasma, respectively, and determined during method validation with a signal-to-noise ratio of 9, an accuracy of 80-120 %, and a precision of ± 20 %.”

Comment:

Page 5, lines 177-178. Authors must provide detailed accuracy and precision data for each level tested. They can present this full data as supplementary material if they want.

Response:

We’ve provided an additional table in the Supplement Material (Table S1) with detailed information on method validation parameters for the determination of PG and G in urine and plasma and refer to this table in section 3 (Results and Discussion).

Comment

Page 6, line 235. Was this statistical comparison between the collection times performed for each group? If so, were significant differences found only for EC? What does the asterisk in the number 0.09 of the control group mean?

Response

For statistical analysis between the collection times was performed for each group separately. We have corrected the paragraph for the sake of clarification.

The asterisk at the number 0.09 of the control group was a typo and thus corrected.   

“The observed increase of the PG levels in EC user was significant for both collection time points (p < 0.01, Mann-Whitney U test). No significant differences were found between the other groups investigated. Moreover, plasma PG increased significantly (Mann-Whitney U test, p < 0.01) throughout the day in EC users with average values of 1.25 µg/mL in the morning and 8.37 µg/mL in the evening (Figure 3). With the exception of the NU group, in which a significant increase of plasma PG from 0.05 µg/mL in the morning to 0.09 µg/mL in the evening was observed, none of the other groups showed an increase in PG levels over day.”

Comment:

Page 7, Table 2. What does this asterisk mean? According to the authors, the level of significance was set at 0.01 and is presented with 3 asterisks. So what does an asterisk mean?

Response:

This was a typo. We have corrected it accordingly.

Comment:

Page 7, Table 2. It's pretty clear that this concentration of PG is an outlier. Why didn't the authors remove this value?

Response:

We agree with the reviewer that this value is unusually high. However, due to the small sample size we decided to keep this value for evaluation since it correlates not only with the calculated PG intake but also with total nicotine equivalents (TNE) (see Figure 5). Thus we considered this value as a reasonable high concentration resulting from high PG intake rather than an outlier.

Comment:

Page 8, lines 252-253. This doesn't seem right. The Kruskal Wallis test does not work that way. It compares each group with the others. There are specific statistical tests to compare just one group with others. Review this.

Response:

For this statistical assessment, EC users were compared with each other group (comparison of two groups) using Mann-Whitney U test. We have added this information to the figure caption for clarity.

“*** p < 0.01 (Mann-Whitney U test, comparison of EC with each other group).

Comment:

Page 8, line 258. Figure 4b is of limited relevance because one cannot "see" the results of groups other than the EC. I advise the authors to find a way to make the data presentation meaningful. For example, authors can try to log their data.

Response:

Thank you for this advice. We kindly refer to Table 3 (urine), in which the descriptive statistics of all groups (mean, median, SD, min, max) is given. We additionally added the reference to the corresponding table in the sentence to clearly refer to these data.

Comment:

Page 8, lines 259. Mann-Whitney?! Shouldn't it be Kruskal Wallis? There are more than two independent groups in this comparison. Correct.

Response:

For this statistical assessment, EC users were compared with each other group (comparison of two groups) using Mann-Whitney U test. Since the p-value was the same for all group comparisons, it was reported only once.

Comment:

Page 8, lines 264. Did the authors use the Kruskal Wallis test to confirm the results from the Mann-Whitney's? To start, why was the MW test performed when there are more than two independent groups?

Response:

MW test was performed to see if there are any significant differences between the control group (non-user) and each of the other groups investigated in this study, except for EC users. Kruskal Wallis test was performed to confirm the results and to see if there are any other significant differences between the other groups.

Comment:

Page 9, lines 274. Where can I see these results? Figure 3 shows the PG plasma concentration on day 1 but there is no table or figure with the PG urinary concentration on day 1.

Response:

We thank the reviewer for this advice. Unfortunately, we did not refer to Figure S3 in Supplementary Material, which shows the urinary and plasma PG data at day -1. This was corrected in the latest version. In addition, we’ve added Table S6 (Supplementary Material) giving the descriptive statistics of PG in plasma and urine on day -1 (U0, B0).

Comment:

Page 9, lines 275-277. Is this a result of the present study? Where are the results that support this claim, that is, where are the results of "real-life uncontrolled conditions"?

Response:

The results mentioned derive from the present study. Immediately after admission to the clinical site, (day ‑1, 5 pm) plasma and urine (12 h urine) samples were collected. These samples can be considered as real-life samples as the subjects were not subject to any restrictions prior to admission to the clinic. As mentioned before, we’ve added Table S6 (Supplementary Material) giving the descriptive statistics of PG in plasma and urine on day -1 (U0, B0). Figure S3 as well as Table S6 in Supplementary Material show urinary and plasma PG data on day -1. We have added these references to the sentence for clarity.

Comment:

Page 9, lines 286. Where did this value come from? Is this the average PG concentration in EC users?

Response:

PG and C concentrations analyzed in urine and plasma were determined in µg/mL. For data evaluation we used the measured PG concentration in µg/mL to calculate the urinary excretion based on the urine volume of the voids. For the sake of clarification we’ve added the unit in brackets under 2.4 (data evaluation).

“Urinary analyte concentrations (µg/mL) were multiplied with the respective 12 h urine volume to receive the amounts of PG and G excreted within 12 h. The appropriate 12 h urine pools were summed up to obtain the amount of the analytes excreted in 24 hours. Evening of the first day until evening of the next day were defined as a 24 h interval (U0 + U1/2, U3 + U4/5, U6 + U7/8). Urinary PG and G excretions were expressed in mg per 24 hours (mg/24h).”

Comment:

Page 9, lines 286. "in accordance"?! Reported levels are nearly 10 times lower than Wurita's. How is this in accordance? The author must explain these values and compare them with other papers other than Wurita's to understand whether these values are in fact typical or not.

Response:

We agree with the reviewer that the reported concentrations seem to be 10 times lower than described in Wurita et al. However, they observed a few high values which is why we reported the range (min-max) of PG levels observed in these subjects. In addition, we performed the Mann-Whitney U test to compare the Wurita et al. (data from Wurita et al. was given) with our own data, with the result that there were no significant differences between these two groups (p > 0.01).

We have rephrased this sentence to make this clear and have additionally addressed the reviewer’s suggestion to add another publication.

“In relation to the excreted volume, PG levels in urine correspond to 672 ± 209 ng/mL, not significantly different (Mann-Whitney U test, p > 0.01) from Wurita et al. that observed PG levels of 5450 ± 9290 ng/mL (mean ± SD, range: 491-41600 ng/mL) in urine of 23 healthy subjects [36]. In addition, if PG was normalized to creatinine (determined as part of the project), the PG level in the control group (NU) was 1.6 ± 0.6 mmol/mol creatinine, which is comparable with the value of 2.3 ± 1.4 mmol/mol creatinine observed by Laitinen et al. {Laitinen, 1995 #150003}.”

Comment:

Page 9, lines 295. Confusing sentence. Is the uptake of PG by EC users any different from the other nicotine-containing products (e.g., CC, HTP, NRT)? I would imagine that the PG concentration is higher in EC users due to the high concentration of PG in those products and not due to differences in the uptake process. Can the authors clarify this?

Response:

We agree with the reviewer that PG concentration is higher in EC users due to the high concentration of PG in those products. The sentence was a bit misleading. We therefore rephrased the sentence for clarity.

“As mentioned before, the EC users show by far the highest PG values, which can be attributed to PG intake from EC use [44,45].

Reviewer 3 Report

Overall, this is a well written paper. The authors observed that 1,2-propylene glycol (PG) levels in urine and plasma in electronic cigarettes (EC) users were much higher compared to users of other nicotine products including combustible cigarettes (CC), heated tobacco products (HTP), oral tobacco (OT), nicotine replacement therapy (NRT), as well as a group of non-nicotine users (NU).  The PG in plasma and urine of vapers significantly correlated with nicotine (plasma) and total nicotine equivalents (urine). Therefore, they proposed that PG can be used as biomarkers of exposure in urine and/or plasma to monitor EC use compliance in exposure assessments. From my opinion, this manuscript can be accepted for publication after addressing the following question:

  1. P6, lines 221-223, the authors wrote “Data demonstrate that G is not elevated in users of EC, neither in plasma nor urine, and therefore cannot differentiate EC specific uptake of G from other nicotine products.” Since the ratio of PG/G is quite different in different EC products, did the authors observe any correlation of the glycerol in blood or plasma with the calculated glycerol intake? It can be very helpful if the authors list the propylene glycol and glycerol amounts in the EC products used in the tests.

Author Response

Response to reviewers

Ref: separations-1360630

Title: 1,2-Propylene glycol: A biomarker of exposure specific to e-cigarette consumption

Journal: Separations

First of all, we would like to thank the reviewers for their helpful comments, which substantially improved the manuscript. We hope that we could address all raised issues. For better traceability we highlighted the added/revised text in red.

Reviewer 3:

Overall, this is a well written paper. The authors observed that 1,2-propylene glycol (PG) levels in urine and plasma in electronic cigarettes (EC) users were much higher compared to users of other nicotine products including combustible cigarettes (CC), heated tobacco products (HTP), oral tobacco (OT), nicotine replacement therapy (NRT), as well as a group of non-nicotine users (NU).  The PG in plasma and urine of vapers significantly correlated with nicotine (plasma) and total nicotine equivalents (urine). Therefore, they proposed that PG can be used as biomarkers of exposure in urine and/or plasma to monitor EC use compliance in exposure assessments. From my opinion, this manuscript can be accepted for publication after addressing the following question:

Comment:

P6, lines 221-223, the authors wrote “Data demonstrate that G is not elevated in users of EC, neither in plasma nor urine, and therefore cannot differentiate EC specific uptake of G from other nicotine products.” Since the ratio of PG/G is quite different in different EC products, did the authors observe any correlation of the glycerol in blood or plasma with the calculated glycerol intake? It can be very helpful if the authors list the propylene glycol and glycerol amounts in the EC products used in the tests.

Response:

We have correlated the calculated glycerol intake with the glycerol levels in urine and plasma, respectively. Unfortunately, we did not see any correlation (Spearman’s rank correlation, r < 0.0). We additionally addressed the propylene glycol and glycerol amounts in the EC products in section 2.1 (Study Design).

“The amount of PG and G in the e-liquids consumed by the subjects ranged from of 50-55 % for PG and 45-50 % for G, respectively.”

Round 2

Reviewer 1 Report

The manuscript titled “1,2-Propylene glycol: A biomarker of exposure specific to e-cigarette consumption” (separations-1360630.R1) has been updated to address some of the raised concerns, but it remains largely unclear how authors came to the added conclusions and limitations. For example, it remains unclear how the value of “50-55% PG and 45-50% G” was determined. Was did determined analytically? Did users report these values? Where they confirmed? Also, authors have now included text suggesting that their method can be utilized as long as 10% PG is present in a given e-liquid. It remains completely unclear how that value was derived and what the real-world implications are. Again, the study was extremely controlled, i.e., participants were only allowed to consume specific foods during the study, so even if authors can show their calculation as to why a limit of 10%PG exists, it remains unclear if such a limit would only remotely hold in a real-world scenario, or if the method would simply find that “everyone” is an e-cig user (compare Figure 3, time 0; does the difference between e-cig users and other tobacco users hold statistically? It looks like there is clear overlap between the e-cig, HTP, NRT, and combustible product user group- it’s very hard to read when all error bars overlap). In that case, PG would not be a suitable biomarker. In summary, the added language does not sufficiently address the raised concern. If authors can back up their PG/G ratios and lay out how the value of 10% PG was derived, the manuscript may come to the conclusion that in a non-real-world environment where no other source of PG is present, PG *may* be an appropriate biomarker for e-cig use, but to claim the latter broadly, the authors would have to show their results in a larger, real-world study. The conclusion that PG is a suitable biomarker of EC use (l418ff.) is not supported by the presented data and authors need to be much more careful in their wording. This reviewer will not rule out that PG *may* be a suitable biomarker and acknowledges the value of such a test, but this absolute conclusion can most certainly not be drawn from the presented data.

Author Response

Response to reviewers

Ref: separations-1360630

Title: 1,2-Propylene glycol: A biomarker of exposure specific to e-cigarette consumption

Journal: Separations

Reviewer 1:

The manuscript titled “1,2-Propylene glycol: A biomarker of exposure specific to e-cigarette consumption” (separations-1360630.R1) has been updated to address some of the raised concerns, but it remains largely unclear how authors came to the added conclusions and limitations. For example, it remains unclear how the value of “50-55% PG and 45-50% G” was determined. Was did determined analytically? Did users report these values? Where they confirmed? Also, authors have now included text suggesting that their method can be utilized as long as 10% PG is present in a given e-liquid. It remains completely unclear how that value was derived and what the real-world implications are. Again, the study was extremely controlled, i.e., participants were only allowed to consume specific foods during the study, so even if authors can show their calculation as to why a limit of 10%PG exists, it remains unclear if such a limit would only remotely hold in a real-world scenario, or if the method would simply find that “everyone” is an e-cig user (compare Figure 3, time 0; does the difference between e-cig users and other tobacco users hold statistically? It looks like there is clear overlap between the e-cig, HTP, NRT, and combustible product user group- it’s very hard to read when all error bars overlap). In that case, PG would not be a suitable biomarker. In summary, the added language does not sufficiently address the raised concern. If authors can back up their PG/G ratios and lay out how the value of 10% PG was derived, the manuscript may come to the conclusion that in a non-real-world environment where no other source of PG is present, PG *may* be an appropriate biomarker for e-cig use, but to claim the latter broadly, the authors would have to show their results in a larger, real-world study. The conclusion that PG is a suitable biomarker of EC use (l418ff.) is not supported by the presented data and authors need to be much more careful in their wording. This reviewer will not rule out that PG *may* be a suitable biomarker and acknowledges the value of such a test, but this absolute conclusion can most certainly not be drawn from the presented data.

With regard to the PG and G content of the e-liquids, subjects were asked to provide information on the e-liquids they consumed in the form of a questionnaire. They either provided information directly about the PG/G content of the e-liquid base if they were self-mixed, or the PG/G content was verified on the basis of the manufacturer’s specification. We have added this information to the manuscript.

These values have not been analytically verified, as we see no reason for deception.  

“This information is based on the manufacturer's specifications of the consumed liquids, or on the self-reported PG/G content of the e-liquid base if the liquids were self-mixed.”

Regarding the 10 % PG as cut-off in order to detect differences between users of EC and other nicotine user groups, we would like to explain the calculation in more detail.

On day -1, we observed 9-27 times higher PG levels in plasma (B0) and 12-32 higher PG levels in urine (U0) in EC user compared to the other groups. Study subjects used e-liquids with an average PG content of 50 %. Assuming that the PG content in the liquid decreases by a factor of 5, which would result in a PG content of 10 %, the observed difference in EC user compared to the other groups would mathematically decrease to approx. 2-5 times higher PG levels in plasma/urine. We could confirm that the results are still significant even when the Mann-Whitney U test was performed with 5-fold lower PG levels in EC users. In order to clarify this, we’ve added the following sentence.

“Theoretical mathematical assessment with 5 times lower PG level in EC users still holds statistical significance substantiating the limit 10 % PG content.”

To address the real world implication of the 10 % PG cut-off, we kindly refer to the discussion, in which we’ve already described that 10 % would be the minimum amount of PG in the liquid to detect any difference between users of EC and other nicotine user groups.

“Another limitation is a minimum amount of presumable ≥ 10 % PG in the e-liquid, to detect a difference between users of EC and other nicotine user groups. However, further investigations are needed in order to verify this cut-off.”

With regard to the raised concerns on real-world implications, we would like to address the following aspects:

  1. Even under uncontrolled conditions, data on day -1 (B0 and U0), levels of PG were significantly elevated in users of EC compared to the control group and all nicotine user groups investigated. As shown before, we’ve verified significance with a fairly hard cut-off for significance, α = 0.01, by Mann-Whitney U test. We’ve provided the p-values of the significance test (Mann-Whitney U test) performed at day -1 for urine in Table 1 and for plasma in Table 2 in this response letter. The results are only significant for EC users (compared to any other group, p-values < 0.01 are marked in red). This significance test has, in our opinion, a higher validity than the visual evaluation of Figure 3, in which mean + SD are shown.

  1. The Box-and-whisker plots of PG on day -1 in plasma (B0) and in urine (U0) were already added to the Supplementary Material to unambiguous present the obvious differences between EC users and the other groups under uncontrolled conditions. You can find this graph in this response (Figure 1). Again, the significance was tested with Mann-Whitney U test, α = 0.01.

  1. We kindly refer to discussion and the conclusion of the manuscript, where we’ve unambiguously added the limitations of this study and stated that, in fact, “larger studies under field conditions are required to support the suitability of PG in plasma or urine as specific biomarker for the use of ECs”.

Discussion: “Larger studies under field conditions are required to support the suitability of PG in plasma or urine as specific biomarker for the use of ECs.”

Conclusion: “Due to the restricted sample size of the current study, we recommend verifying these results under field conditions.”

  1. In addition, we have softened the wording by describing PG as “potential” biomarker on condition that results need to be verified under field conditions.

“The observed dose-response relationship between urinary and plasma PG and intensity of vaping (daily consumption and nicotine uptake) emphasizes the suitability of PG as a potential biomarker of EC use. Due to the restricted sample size of the current study, we recommend verifying these results under field conditions.”

As suggested, we’ve addressed information on the PG/G ratio and the value of 10 % PG cut-off in order to detect differences between users of EC and other nicotine user groups. Considering the aspects mentioned above, we have clearly demonstrated that the conclusion drawn is statistically supported and reasonable.